# Cross-partisan discussions reduced political polarization between UK voters, but less so when they disagreed

Jona F. de Jong [1] ✉

Can brief, unmoderated cross-partisan discussions reduce political polarization, even when partisans disagree on the issue under discussion? This article reports results from an experiment that matched 582 UK Labour and Conservative party voters for a ten-minute, unmoderated chat discussion about a contentious issue in a wait-list control design. Issue disagreement between discussing partisans randomly varied, and was visible throughout the discussion. Results show that after the discussion, out-partisan sympathy and willingness to have cross-partisan discussions increased. There was no statistically significant effect on opinions. The effect on sympathy was lower when partisans' issue opinions were further apart. Treatment effects correlate with reported experiences of perspective-getting, inclusion in the discussion, commonality and reduced misperceptions. Conservative respondents about to discuss immigration softened their views just prior to the discussion.

[1] European University Institute, Via della Badia dei Roccettini 9, 50014 Fiesole, FI, Italy.  ✉email: jona.dejong@eui.eu

In several countries around the world, animosity between citizens with different political orientations has risen[1,2], affecting social relations [3–5] and the ability to address complex policy issues[6,7]. Scholars looking for interventions to reduce political polarization often point to the alleged salutary effects of cross-partisan discussions on political issues[8–13], but discussion opportunities are limited in everyday life. Individual social environments tend towards political homogeneity[14–16]. In more heterogeneous environments, citizens often avoid discussing politics[17,18]. Online, people can be exposed to political heterogeneity[19,20], but mere exposure does not seem to reduce polarization[21,22].

Researchers have thus started to study the depolarizing effect of cross-partisan discussions between strangers. Recent experimental studies show that these discussions can reduce political polarization[23–27]. However, little is known about their impact when partisans disagree on the issues under discussion. In real life, citizens often encounter polarizing messages from the media, political campaigns or from other citizens in online environments[19,20,28–30], and it is in these contexts that polarized thinking can become activated[31]. Thus, interventions to polarization ideally work even in an environment that gives free rein to the issue disagreement that partly drives inter-partisan animosity[32,33]. Another, related, open question concerns mechanisms: what is it about these discussions that make people increase their sympathy for the other side?

In this article, I run an experiment on a chat platform that matches 582 UK Labour and Conservative voters for a 10-minute, anonymous, cross-partisan discussion on immigration or redistribution. The study tests the depolarizing effect of cross-partisan discussions in a setting that makes issue disagreement visible, where it exists, by showing discussing partisans each other's party and issue opinion throughout the discussion. In addition, discussing partisans are not provided with discussion tips, nor information about the discussion topic or ample time to discuss, in order to avoid potentially softening disagreement. Random assignment of discussion partners—and thus their issue opinion —allows for a test of whether any depolarizing effect of the discussion depends on issue agreement. The study also explores several hypothesized mechanisms that may explain any depolarizing effect of the discussion. The use of a wait list control design allows for a test of mechanisms while avoiding post-treatment bias. In addition, the study provides real-world observational evidence of the depolarizing effects of the discussion, and the scalability of the intervention[13]. Finally, it tests whether respondents soften their views in anticipation of the discussion. Previous literature shows that people can engage in this behavior in their social networks to avoid political conflict[17] but there is little knowledge on whether this behavior also applies to discussions with strangers.

From existing literature, it is not obvious that a brief, unmoderated discussions would reduce political polarization. Studies on cross-partisan interactions generally take inspiration from two theoretical paradigms, neither of which fits perfectly with the brief, unmoderated and potentially divisive discussions of interest in this study. Work on reducing affective polarization means to increase inter-partisan sympathy and reduce social distance[1,12]. It usually builds on intergroup contact theory, which holds that under conditions of equal status, common goals, cooperation and institutional support, cross-group contact can lower prejudice[34]. However, contact theory implies a need for intimate, sustained contact[35–37] and canonical recent studies— some of which find null effects or modest effects—look at interventions that take weeks or months[38,39]. In addition, discussions can by definition become competitive[40], which does not sit well with the collaborative spirit of contact theory[41]. Scholars looking

to reducing ideological polarization aim to moderate opinions. They often draw from the literature on deliberation. Similar to contact studies, deliberation exercises are heavily moderated and take multiple hours or days[23]. Though substantive disagreement is inherent in deliberation exercises[42], it is softened by information provision or discussion moderation. Furthermore, while deliberation can change opinions, it may not do so when the partisanship of fellow discussants is known. It also remains an open question whether a brief, unmoderated, one-on-one discussion has sufficient deliberative qualities to change opinions [43] (though see ref. [27].) Thus, the intervention of interest in this study has some features of deliberation and contact, but does not sit well with either, and neither theory gives clear-cut predictions as to whether it will reduce polarization.

Rather, a rapidly growing and largely US-based literature on the micro-level dynamics of affective polarization in interactive settings[13,44] points to a series of mechanisms that (1) have been associated with a reduction of polarization and (2) could conceivably naturally arise in a brief discussion. In particular, much recent work has focused on the relationship between polarization and misperceptions about out-partisans. Several studies show that people consistently misperceive out-partisans' policy positions, social composition, ideological extremity, political engagement and levels of hostility towards their own group[45–51]. Providing respondents, in experiments, with accurate information about out-partisans' policy positions, group composition or hostility levels, generally lowers their hostility towards out-partisans. Discussions could be one way in which these stereotypical views are corrected in real life, for instance by making people realize that out-partisans are more moderate, less politically engaged and more civil than they expected. Issue agreement may play a crucial role here. As people overestimate the ideological extremity and policy positions of out-partisans[33,46,48], a discussion will likely expose many participants to out-partisans who, unexpectedly, agree with them. Indeed, this dynamic may be partly driving the salutary findings in recent cross-partisan discussion studies[25,27].

There are a number of additional ways in which discussions could reduce both affective and ideological polarization. Recent experimental studies use prompts and exercises in real or imagined inter-partisan settings, which both effectively reduced polarization, and which could naturally occur in discussions. The experiments reduced affective polarization when they made people feel invited to voice their opinion, made them realize that they have something in common with out-partisans, and exposed them to new perspectives[52–54]. Opinions were changed when they asked people to justify their opinion in front of others, exposed them to persuasive arguments, and had them engage in perspective-taking exercises[55–60]. One or multiple of these behaviors could naturally occur in a discussion setting and cause depolarization, though for ideological polarization, opinions are hard to change especially when people are presented with counter-attitudinal information[61].

Brief discussions may also increase willingness to discuss politics with political opposites. Some studies show that people prefer to not have discussions with out-partisans[62], especially when they expect disagreement[17]. However, this preference could be based on similar misperceptions that drive out-partisan animosity: people may expect a contentious discussion with an uncivil out-partisan. Following a similar logic as above, a positive discussion experience may make them more likely to engage in cross-partisan discussions in general.

Finally, there is the question of whether disagreement in the discussion affects the effects hypothesized above. Past work sees exposure to different views as vital for political moderation[11,42,63]. A recent study finds that encouraging partisans to discuss why they do not identify with the out-party, does not reduce affective

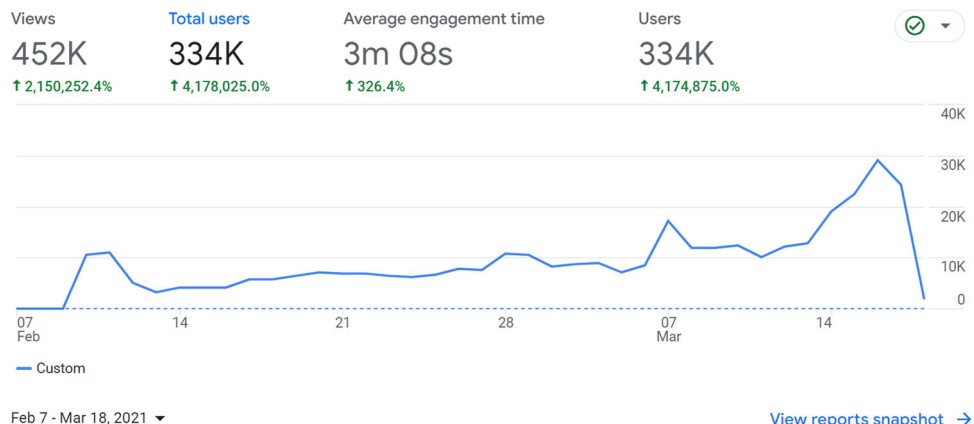

**Fig. 1 Number of visits to discussion platform.** The figure shows the number of clicks on a link that led to the discussion platform in the 5 weeks running up to the Dutch 2021 parliamentary elections. A total of 334.000 users visited the platform, where they spent 3 min on average. Not all users were matched to have a discussion. The number includes multiple visits by the same user. Source: Google Analytics.

polarization[26]. Something similar may happen when opinions are too far apart: discussants will focus on areas of disagreement, the discussion becomes competitive, and polarization is not reduced. At the same time, several of the mechanisms mentioned above could be more likely to occur when partisans disagree: there may be more perspectives to be gotten and more arguments to be voiced. An important additional question here is what will happen without discussion tips or moderation: will discussing partisans attempt to find common ground, or air differences?

## Methods
The experiment was run on a chat platform developed by Civinc, a Netherlands-based NGO. Before conducting the controlled experiment which will be described in this section, the platform was made publicly available in the Netherlands in the 5 weeks before the 2021 parliamentary election. Figure 1 shows that during these weeks, the link to the platform was clicked >300,000 times. This suggests that the platform is scalable. Note that this exercise in the Netherlands was not set up as an experiment, and most data was deliberately not saved. Supplementary Fig. 1 and Supplementary Table 1 show observational results from the little data that was collected, which suggest that these discussions increased out-partisan sympathy. All results reported in the main paper are from the online experiment in the UK, which formally tests whether cross-partisan discussions reduce polarization, while the exercise in the Netherlands shows that the same platform has already been widely implemented in society.

**Sample, recruitment and exclusion**. The controlled experiment was approved by the Ethics Committee of the European University Institute on 13 December 2021. It was pre-registered on June 11th, 2022, before any data collection took place (https://osf.io/q4hjd/). The experiment took place in the weeks after June 11th, in the UK. The UK was selected because of its relatively high levels of political polarization[64]. Respondents were recruited on the online crowd work platform Prolific. They received € 0.27 for the pre-screener (2 min), € 2 for the main experiment (18 min) and € 0.67 for the follow-up survey (5 min). The pre-screener and follow-up surveys took place in Qualtrics; the experiment took place on the discussion platform developed by Civinc, an external programmer and the researcher. All respondents provided informed consent.

A total of 2279 people took the pre-screener. The survey asked about demographics and politics, and took an average of 3 minutes to complete. I asked opinions on five issue statements, and selected the two with the highest average disagreement

between Labour and Conservative voters to serve as discussion topics (immigration and redistribution). Eligibility to partake in the discussion experiment was limited to respondents who indicated to vote Labour or Conservative in the last election, have the UK nationality, have command of the English language and are willing to return for 'a brief, anonymous discussion on a novel chat platform'. This left 1523 respondents. Each respondent was asked to provide a time slot on which they could return for the experiment. Those who did not were dropped, and I randomly dropped Labour voters to invite a balanced sample of Labour and Conservative voters to come to the platform in each wave. A total of 1090 respondents were invited to take part in the experiment. I sent the invitation through a private message in Prolific's internal messaging system. Invitees were asked to mark the time slot in their agenda. They were also instructed that participation and payment were conditional upon arriving exactly on time. A few hours before each time slot, I make the study available on Prolific to invited participants. Thirty minutes before each time slot, I sent participants a final message reminding them that the study was about to start.

777 participants, around 60% of those invited, showed up. An algorithm immediately randomly assigned them to the control condition or the treatment condition (see Fig. 2). Participants who were too late or too early to be matched were excluded. Others were excluded, but paid, because they could not find a match. This was due to an imbalance in partisanship within waves. Though invitations to each wave were balanced by partisanship, attendance was not always balanced. As instructed, 78 participants reported to me by private message that they could not find a match, after having waited for at least 5 min. The data show how long participants spent on the survey, and whether they entered the matching process, allowing me to see whether they actually waited 5 min. Some participants did not contact me, but instead re-entered the platform. In this case, I took out both these participants and their discussion partner. A few others reported a technical glitch: the platform froze, or they were thrown out. If this happened before the treatment, I took them out. If it happened during the treatment, I also took out their discussion partner. I left them in if it happened after the treatment. Of 777 respondents who showed up for the follow-up study, $N = 582$ completed the experiment. Supplementary Notes 2. contains more details on recruitment, survey questions, manipulation checks, attrition and sample demographics.

**Measures**. I use two measures for affect, one for opinions and one for willingness to have out-partisan discussions. For affective

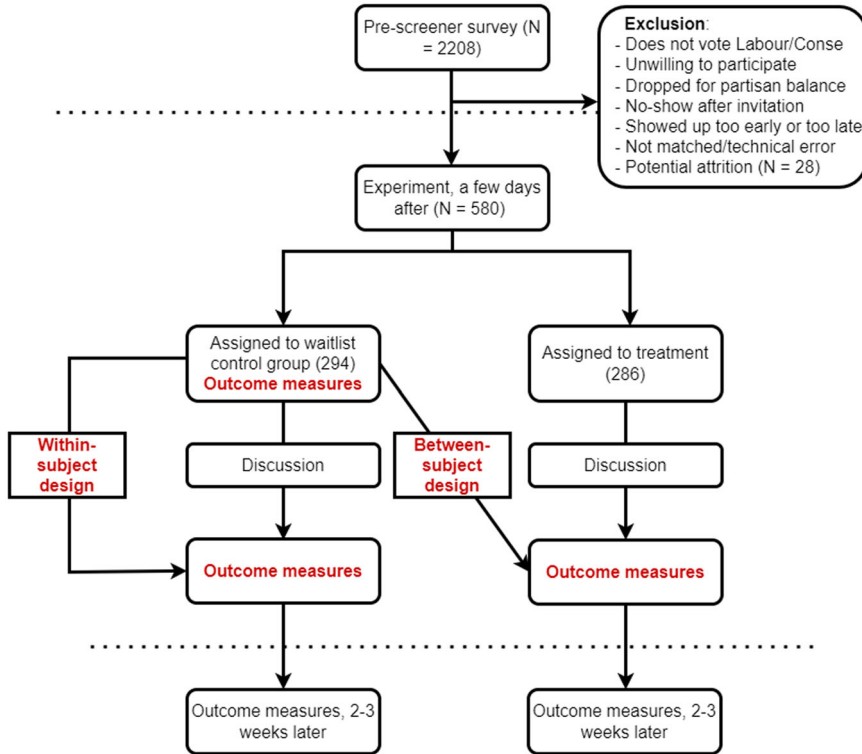

**Fig. 2 Experimental flow.** Respondents take a pre-screener survey a few days before the experiment. Upon returning for the experiment, those in the treatment group are matched to an out-partisan and asked to have a 10-min discussion, after which they fill in the outcome measures. The control group goes through the same flow, but fills in the outcome measures prior to the discussion. At this time, they are unaware that they are meant to discuss politics with someone from a different political party. The between-design compares pre-discussion measures in the control group to post-discussion measures in the treatment group. The within-design compares the change in the outcome variables in the control group before and after the discussion. All results presented in the paper are from the between-design.

polarization, I use a feeling thermometer and a social distance item, both of which are widely used in the literature[48]. Opinions are measured on a 5-point Likert scale that ranges from 'strongly agree' to 'strongly disagree'. To get more insight into discussion mechanisms, I first ask respondents to write a few sentences about how they experienced the discussion. Then, I ask several closed-ended statements about the discussion, which are meant to tap into experiences of perspective-getting, inclusion in the discussion, commonality and quality of arguments. Supplementary section B4 shows all survey questions.

**Analysis**. I conducted a pre-registered power analysis with the R-package 'R-power' for the main effect of the discussions on out-partisan sympathy, and for the test of the discussion effect by baseline issue distance. Final sample sizes were determined in order to detect effects of at least 0.35 standard deviations, found in similar studies[25,26]. Power was set to 80%, and the alpha-level was 0.05. All analyses described below have been pre-registered unless indicated otherwise.

To identify the effect of the discussion on depolarization, I use the between-subject design shown in Fig. 2. I estimate the immediate effect of the discussion with an OLS regression model that controls for baseline covariates (gender, education, ethnicity, political interest, outparty closeness, partisan strength) and clusters standard errors at the discussion level. To measure persistence of treatment effects, participants are re-surveyed 2–3 weeks after the treatment. At this point, the control group has also been treated. Thus, participants in both control and treatment groups could have been influenced by other events between the experiment and the follow-up study, calling for a

cautious interpretation of these results. I again use a between-subject design, comparing polarization levels of the treated 2–3 weeks after the treatment, to those in the waitlist control group. I replicate these analyses with a within-subject design in which I conduct paired t-tests on the control group, comparing their answers to the items after the discussion with just before.

Several exploratory analyses were not pre-registered. A close reading of the chats conversations revealed that many respondents reported reduced misperceptions, which were subsequently included in the analysis of mechanisms. It also revealed a clear difference between dyads who disagreed but tried to find common ground, and dyads who only disagreed. This too was then included in the analysis of mechanisms. The probing of the effect of the treatment by issue distance, by opinion strength was also not pre-registered.

**Statistics and reproducibility**. The dataset contains $N = 582$ observations. Numbers of observations differ slightly between models because some respondents did not fill in certain outcomes, and about ten people could not fill in their opinion just before the discussion. All analyses use the same model (which I will refer to as the basic model), which varies slightly depending on moderators and mechanisms of interest. This model, used to make Fig. 3, uses linear regressions to estimate treatment effects. I regress each outcome on an indicator for treatment, with the wait-list control group as the comparison. I include baseline covariates (education, gender, ethnicity, political interest, outparty closeness, partisan strength) to increase precision, and cluster standard errors at the dyad level. All outcome variables are normally distributed and all tests were two-tailed. I use an alpha-

level of 0.05 to test for statistical significance. Interpretations of certain null results are accompanied by equivalence tests.

To make panels a, b, and c in Fig. 4, I use this same basic model and interact the treatment with a continuous measure of baseline opinion distance. The measure captures the difference between dyads' baseline opinions, and ranges from 0 (complete agreement) to 4 (complete disagreement). I additionally control for a respondent's own baseline opinion. To make panels d, e and f of the same figure, I subset the model by a measure of baseline opinion strength, and now no longer control for a respondent's baseline opinion. The opinion strength variable consists of three values: (1) strong opinions (strongly agree, strongly disagree), (2) moderate opinions (agree, disagree), or a (3) neutral position (neutral). To formally test whether treatment effects by issue distance, differed between respondents with a strong or moderate initial opinion, I run a three-way interaction model, interacting the treatment, the dummy for baseline opinion distance, and a respondent's baseline opinion strength (strong or moderate).

To make Fig. 5, I run the basic model, but do not have a measure of wait list opinions in the control group. Thus, to measure opinion change, I compare treatment effects to baseline opinions of the control group in the pre-screener. These results are displayed in the third row of the Figure. To measure whether respondents hide their views, I compare opinions in the control group in the pre-screener, to opinions in the treatment group just before the discussion (first row). To measure whether views revert back, I compare opinions in the control group just before the discussion, to opinions in the treatment group just after (second row). For the effects by partisanship, I do not cluster SEs at the dyad level, because partisans did not discuss with their in-group. I run interaction models to formally test effects by partisanship.

For Figs. 6 and 7, I run the basic model and include an interaction for the expressed disagreement variable, and the various mechanisms. The data and code to reproduce all of the analyses can be found under 'data availability' and 'code availability'.

### The experiment
*Wait-list control design.* Figure 2 shows the wait-list control design chosen for this experiment. When participants entered the discussion platform, they were randomly assigned to a treatment condition or to a control condition. In the treatment condition, participants proceeded to have a cross-partisan discussion as soon as they entered the platform. In the control condition, participants did not immediately have a discussion. Instead, they were asked to fill in the outcome measures first. Importantly, at this point, they had not yet been told that the discussion would be about politics, or that they would discuss with someone from a different party. The design establishes treatment effects by comparing outcome measures in the wait-list control group, to outcome measures in the treatment group, elicited a few minutes later, right after their discussion. A wait-list control design has two advantages. First, it can be used when all participants need to receive the treatment. In this case, the NGO that owns the platform requested that each participant would have a cross-partisan discussion. Second, it makes possible the testing of mechanisms with the use of post-treatment variables, but without post-treatment bias, because the control group also gets treated. Thus, I can compare, for example, dyads in the control group who will engage in certain discussion behavior, to dyads in the treatment group who did engage in this behavior. Limitations of this design are addressed in the discussion section of this article.

*Discussion treatment.* The discussion treatment went as follows. After having selected their party, participants were told that they were meant to have a 10-minute discussion with someone from a different political party. An algorithm then matched them to another respondent from the opposite party (Labour or Conservative) who was simultaneously present on the platform. Both participants could see the party of their match and were given one of two randomly assigned topics to discuss: immigration or redistribution. After being presented with the statement, participants were asked to give their opinion. Importantly, they were told that their opinion would be visible to their partner during the discussion. It was not an option to not give their opinion. This allows me to see whether participants hid their views from one another, by comparing privately provided opinions in the pre-screener, to publicly provided opinions just before the discussion. Throughout the discussion, both participants could see the party and issue position of their match. After the discussion, participants were presented with the outcome measures. Outcome measures were asked again a few weeks after the experiment, to see whether treatment effects persisted. The follow-up survey was only visible to those who completed the discussion, and was unobtrusively titled 'brief survey on politics and society'. This discussion setting deliberately minimized researcher involvement. I provided only a statement to discuss and a time limit. Apart from that, participants could have a free-flowing discussion without moderation or guidance.

**Reporting summary.** Further information on research design is available in the Nature Portfolio Reporting Summary linked to this article.

## Results
**Participant demographics and attrition.** Table 1 shows characteristics of the final sample, compared to all who completed the pre-screener survey, and to respondents in the British Election Study Wave 23, which was fielded at approximately the same time as this study. Participants in the final sample reported a mean age of 44, with an SD of 14. 57% of the sample self-identified as women; 43% as men. Supplementary Fig. 2 shows that besides higher-educated participants being slightly more likely to have made it from the pre-screener into the experiment, there is no statistically significant difference on baseline policy attitudes, closeness towards parties, and ideology. Compared to the BES, the sample as a whole is slightly more educated and younger, and consists of stronger partisans. There was little attrition: 12 participants waited a few minutes for a match and then returned their submission without contacting me. It is likely that they simply could not find a match. Another 15 respondents were matched, sometimes sent a few messages to their discussion partner, but then left the discussion without contacting me. I checked all the discussions and all but two seem normal. Supplementary Fig. 3 assumes that these 27 respondents attritted, and shows that attrition is unrelated to treatment status or baseline covariates.

**Affective polarization.** All analyses in this sections were preregistered, unless otherwise indicated. I first test the effect of the discussion on out-partisan sympathy, willingness to have out-partisan friends, and willingness to discuss with out-partisans. Figure 3 reports the results from regression analyses of the treatment effects. Outcome variables are measured on 11-point scales, and treatment effects report mean differences between the treatment group and the control group. Positive values denote hypothesized reductions in polarization. There is an immediate depolarizing effect of the discussion. Respondents assigned to the discussion treatment exhibit higher levels of out-partisan sympathy (t(296) = 6.7, $P < 0.001$, ATE = 1.19, $d = 0.43$, 95%

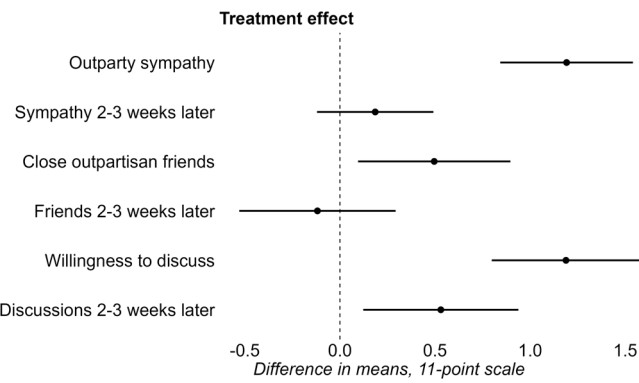

**Fig. 3 After the discussion respondents show decreased levels of affective polarization and increased willingness to have discussions. Two to three weeks after the discussion, only the effect on willingness to discuss persists.** Point estimates and 95% Confidence Intervals (CIs) show the effect of the discussion treatment, on an 11-point scale. The estimates show differences in means between the treatment group and the wait-list control group. Positive values on the x-axis denote depolarization (increased sympathy, increases in feeling comfortable with having close out-partisan friends, increases in feeling comfortable to discuss politics with an out-partisan). Estimates are calculated using a linear regression model that controls for baseline covariates (gender, education, ethnicity, political interest, outparty closeness, partisan strength) and clusters SEs at the dyad level. After the discussion, respondents exhibit higher levels of out-partisan sympathy (t(296) = 6.7, P < 0.001, ATE = 1.19, d = 0.43, 95% Confidence Interval (CI) = (0.84–1.54)), feeling comfortable with having out-partisan friends (t(296) = 2.43, P = 0.02, ATE = 0.5, d = 0.17, 95% CI = (0.09–0.9)) and willingness to have political discussions with out-partisans (t(296) = 5.97, P < 0.001, ATE = 1.19, d = 0.47, 95% CI = (0.8–1.58)). I find no statistically significant effect of the discussion on sympathy (t(557) = 1.19, P = 0.24, ATE = 0.19, d = 0.01, 95% CI = (−0.12–0.49)) or friends (t(557) = -0.56, P = 0.57, ATE = −0.12, d = -0.05, 95% CI = (−0.53–0.29)) 2–3 weeks later. Effects on discussions do persist (t(557) = 2.55, P = 0.01, ATE = 0.53, d = 0.21, 95% CI (0.12, 0.94)), under the assumption that observed effects are not influenced by any events in the 2–3 week period after the treatment.

Confidence Interval (CI) = (0.84–1.54)), of feeling comfortable with having out-partisan friends (t(296) = 2.43, P = 0.02, ATE = 0.5, d = 0.17, 95% CI = (0.09–0.9)) and of willingness to have political discussions with out-partisans (t(296) = 5.97, P < 0.001, ATE = 1.19, d = 0.47, 95% CI = (0.8–1.58)) compared to the waitlist control group. To see whether effects persisted, I compare outcome measures of the treatment group 2–3 weeks after the experiment, to those in the wait-list control group. I limit the sample to those in the control group who responded to the follow-up survey. This analysis calls for cautious interpretation of results, as it requires the additional assumption that observed effects are not influenced by other events in the weeks between which the two groups were measured. I find no statistically significant effect of the discussion on sympathy (t(557) = 1.19, P = 0.24, ATE = 0.19, d = 0.01, 95% CI = (−0.12–0.49)) or friends (t(557) = −0.56, P = 0.57, ATE = −0.12, d = −0.05, 95% CI = (−0.53–0.29)) 2–3 weeks later. In contrast, the effect on willingness to have a discussion with someone from the other side of the political spectrum (t(557) = 2.55, P = 0.01, ATE = 0.53, d = 0.21, 95% CI (0.12, 0.94)) does persist.

Supplementary Table 3 shows the regression tables. Supplementary Fig. 4 shows that results are similar when using the within-individual design. Supplementary Fig. 5 tests for demand effects, by using feelings towards in-party voters, Green voters and Liberal-Democrat voters as placebo outcome measures. Sympathy towards these groups should be largely unaffected by the discussion. This is

indeed what the figure shows, with the exception of a small increase in sympathy towards Green party voters among respondents who vote for the Conservative party. Finally, Supplementary Table 4 shows no partisan asymmetry in treatment effects.

*Affective polarization by issue disagreement.* Does the depolarizing effect of the discussion depend on issue disagreement between discussing partisans? The design allows for a test of this question. Baseline issue distance depends on a respondent's baseline opinion and that of their partner. By controlling for respondents' baseline opinions, variation in issue distance only comes from their partner's opinion. As partners are randomly assigned, so is issue distance. Supplementary Table 5 shows that baseline issue distance is uncorrelated with pre-treatment covariates and treatment status.

In Fig. 4, panels a, b and c, I run the same models as in Fig. 3, controlling for baseline opinions, and including an interaction term denoting baseline issue distance. Opinions were measured on a 5-point Likert scale. The issue distance variable thus runs from 0 (complete agreement) to 4 (complete disagreement). Figure 4 shows that the effect of the treatment on sympathy depends on issue distance: as issue distance increases, the effect of the treatment decreases (t(269) = −2.02, P = 0.045, ATE = −0.32, 95% CI = (−0.63−−0.006)). In the plot, the treatment effect appears to vanish above a distance of about 2.5, but I cannot reject the presence of a small effect with an equivalence test (t(136) = −1.6, P = 0.06). No similar statistically significant effect is found for the other outcome variables (Friends: (t(269) = −0.07, P = 0.95, ATE = −0.01, 95% CI = (−0.42–0.4))); Discussion: (t(269) = −0.65, P = 0.52, ATE = −0.13, 95% CI = (−0.54–0.27)). Issue agreement appears as an important condition for cross-partisan discussions to reduce animosity.

I further probe the effect on sympathy in Fig. 4d–f. This is an additional analysis, that was not preregistered. Using the same interaction models, I conduct separate analyses for respondents with strong baseline opinions ('strongly agree' or 'strongly disagree'), moderate opinions ('agree' or 'disagree'), and the neutral position. This analysis means to account for the fact that respondents with stronger baseline opinions can disagree to a greater extent (i.e. higher potential issue distance and thus higher potential treatment dosage), than those with moderate opinions. Results are suggestive, but inconclusive. Panels d, e and f of Fig. 4 show that for participants with strong baseline opinions (N = 187), the effect of the discussion on sympathy depends significantly on baseline disagreement (t(146) = −2.08, P = 0.04, CATE = −0.5, 95% CI = (−0.99–0.02)). The same interaction term is not statistically significant for those with moderate opinions (N = 287), (t(204) = −1.29, P = 0.2, CATE = −0.32, 95% CI = (−0.83–0.18)), or neutral opinions (N = 93) (t(84) = 0.2, P = 0.85, CATE = 0.17), 95% CI = (−1.65–1.99). Supplementary Tables 8 and 9 show that these results are robust to using a dummy version of the distance variable, which takes the value (1) when one respondent (strongly) agrees, and the other (strongly) disagrees with the statement, and (0) otherwise. The differential effects by baseline opinion strength just lose statistical significance when formally testing them with a three-way interaction model in Supplementary Table 10 (t(265) = −1.66, P = 0.1, CATE = −1.25, 95% CI = (−2.72–0.23)). I thus find no statistically significant difference of the effect of the discussion by issue distance, between respondents with strong and moderate baseline issue positions.

**Opinion change**. Did the discussion moderate opinions? Respondents provided their opinions at four points in time. In the pre-screener (t0), just before the start of the discussion (t1), right after the discussion (t2) and 2–3 weeks later (t3). Recall that right

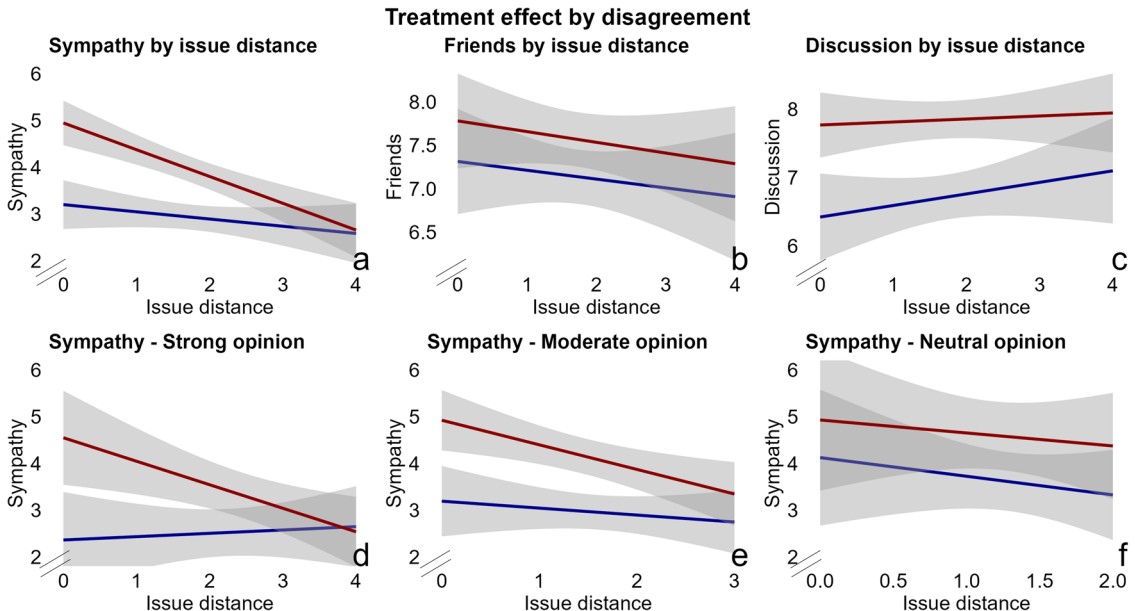

**Fig. 4 The effect of the discussion on out-partisan sympathy depends on baseline issue distance to one's conversation partner.** Regression lines and 95% CIs (shading) are shown for the control group (blue) and the treatment group (red). Models show outcomes by issue distance, presented on the x-axis. I report interaction terms because I hypothesize that treatment effects depend on issue distance. The models interact the treatment with a continuous measure of baseline issue distance (0–4), controls for baseline covariates (gender, education, ethnicity, political interest, outparty closeness, partisan strength and a respondent's initial position), and clusters SEs at the dyad level. To make panels (**d**–**f**) I do not control for a respondent's baseline opinion, but subset by baseline opinion strength. Panels **a**–**c** Show that the effect of the discussion on sympathy depends on discussion distance ((**a**) (t(269) = −2.02, P = 0.045, ATE = −0.32, 95% CI = (−0.63−−0.006))). There is no statistically significant effect of the treatment by discussion distance for the friends variable ((**b**) (Friends: (t(269) = −0.07, P = 0.95, ATE = −0.01, 95% CI = (−0.42–0.4))) or the discussion variable ((**c**) (t(269) = −0.65, P = 0.52, ATE = −0.13, 95% CI = (−0.54–0.27))). For participants with strong baseline opinions, the effect of the discussion on sympathy depends significantly on baseline disagreement (panel (**d**) (t(146) = −2.08, P = 0.04, CATE = −0.5, 95% CI = (−0.99-0.02))). The same effect is not statistically significant for those with moderate opinions ((**e**) (t(204) = −1.29, P = 0.2, CATE = −0.32, 95% CI = (−0.83-0.18))) or those with neutral opinions ((**f**) (t(84) = 0.2, P = 0.85, CATE = 0.17), 95% CI = (−1.65–1.99))). However, the difference between those between strong and moderate opinions is not statistically significant (t(265) = −1.66, P = 0.1, CATE = −1.25, 95% CI = (−2.72–0.23)).

before the discussion, respondents were told that their opinion would be visible to their discussion partner, and they did not have the option to not provide their opinion. I re-code the opinion variables so that higher values indicate agreement with the out-party's traditional opinion. Thus, higher values for Labour voters stand for conservative immigration or redistribution attitudes, and higher values for Conservative voters stand for progressive immigration or redistribution attitudes. Opinion depolarization means an average shift towards the position of the out-party.

To estimate the effect of the discussion on opinion change, I use a slightly different design than for the other variables. As issue opinions were not asked to the wait-list control group right before their discussion, I compare the mean of the control group at t0 to the mean of the treatment group at t2. The only difference from the design used for the other variables, is that baseline opinions were measured a few days before the discussion, instead of a few minutes before the discussion. Supplementary Table 11 shows that both baseline opinions, and opinions right before the discussion, were statistically identical across treatment and control. All regression models are in Supplementary Tables 12–15.

Figure 5 shows opinion moderation at three points in time. First, I test whether opinions moderated significantly between t0 and t1, that is: whether respondents hid their views when asked to provide them publicly. The top row of Fig. 5 shows that respondents indeed moderated their views in anticipation of the discussion (t(296) = 2.13, P = 0.03, ATE = 0.19, 95% CI = (0.01-0.36)). Subgroup analysis shows that the effect is concentrated among Conservative voters (t(280) = 2.85, P = 0.005, CATE = 0.37, 95% CI = (0.12–0.63)), while Labour voters do not display

this behavior (t(271) = −0.2, P = 0.84, CATE = −0.02, 95% CI = (−0.26–0.12)). Supplementary Table 13 interacts the treatment with party membership and indeed shows that hiding views varies significantly by partisanship (t(296) = −2.04, P = 0.04, CATE = −0.38, 95% CI = (−0.75–0.01)). In addition, based on an equivalence test, I can reject the presence of a meaningful effect for Labour voters (t(257) = −2, P = 0.02). The pre-discussion moderation is open to multiple interpretations, but I suspect that Conservative respondents engage in this behavior to avoid a contentious discussion. Supplementary Table 16 shows a further breakdown by discussion topic, which was randomly assigned. A three-way interaction shows that Conservative voters about to discuss immigration moderate their views to a larger extent than those about to discuss redistribution, though the P-value of the interaction term is only borderline significant (t(296) = 1.87, P = 0.06, ATE = 0.63, 95% CI = (−0.03–1.3)). I will return to this finding in more detail in the discussion. Finally, there is no statistically significant evidence that respondents revert back to their original position after the treatment (t1 − t2) (t(296) = −1.19, P = 0.23, ATE = −0.09, 95% CI = (−0.24–0.06)), nor a statistically significant effect of the treatment on opinion change (t0 − t2) (t(296) = 0.39, P = 0.69, ATE = 0.03, 95% CI = (−0.13–0.2)).

In Supplementary Table 17, I replicate the analysis with a within-subject design, using all participants. The results do not change. In Supplementary Tables 18–22, I run a placebo test to make plausible that results are not driven by regression to the mean. I compare opinions on all five issues asked to the control group in the pre-screener, to opinions in the treatment group in

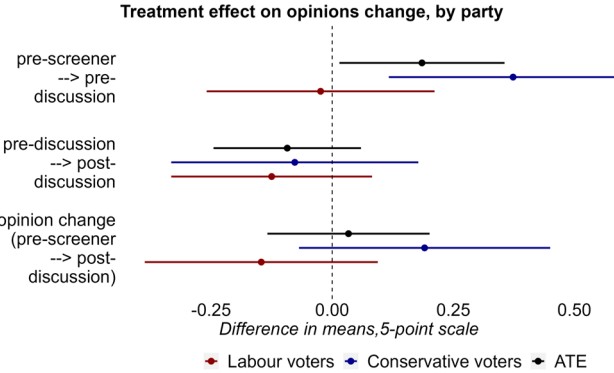

**Fig. 5 Conservative voters hide their views in anticipation of the discussion. There is no statistically significant effect of the discussion on opinion change.** Point estimates and 95% CIs show the effect of the discussion treatment on a 5-point Likert scale. The pre-screener − > pre-discussion results show differences in means between the control group in the pre-screener, and the treatment group right before the discussion. The pre-discussion − > post-discussion results show difference in means between the control group just before the discussion, and the treatment group right after the discussion. Finally, the pre-screener − > post-discussion results show difference in means between the control group in the pre-screener, and the treatment group right after the discussion. Positive values on the x-axis denote depolarization, measured as reduced distance to the out-party's general position. Estimates are calculated using a linear regression model that controls for baseline covariates (gender, education, ethnicity, political interest, outparty closeness, partisan strength). Effects are shown for all respondents (black lines), and for Labour (blue lines) and Conservative (red) lines voters separately. For the ATE, I clusters SEs at the dyad level. For the subgroup analyses, I report traditional SEs, as respondents from the same party did not interact. Results show that respondents moderated their views before the discussion (t(296) = 2.13, P = 0.03, ATE = 0.19, 95% CI = (0.01-0.36)). The effect is concentrated among Conservative voters (t(280) = 2.85, P = 0.005, CATE = 0.37, 95% CI = (0.12-0.63)); I find no statistically significant effect on Labour voters (t(271) = −0.2, P = 0.84, CATE = −0.02, 95% CI = (−0.26-0.12)). There is also no statistically significant effect on opinion change between right before, and right after the discussion (t(296) = −1.19, P = 0.23, ATE = −0.09, 95% CI = (−0.24-0.06)), neither for Conservative voters (t(280) = −0.6, P = 0.55, CATE = −0.08, 95% CI = (−0.33-0.18)) nor for Labour voters (t(271) = −1.18, P = 0.24, CATE = −0.12, 95% CI = (−0.33-0.08)). There is also no statistically significant effect of the discussion on opinion change (t(296) = 0.39, P = 0.69, ATE = 0.03, 95% CI = (−0.13-0.2)), neither for Conservative voters (t(280) = 1.44, P = 0.15, CATE = 0.19, 95% CI = (−0.07-0.45)) nor for Labour voters (t(271) = −1.19, P = 0.23, CATE = −0.15, 95% CI = (−0.39-0.09)).

the follow-up survey, using the same models as above, a few weeks after the treatment. I find no evidence for any shifts in opinions, which one would expect if results were indeed driven by regression to the mean.

**Mechanisms**. In this final section, I use the chats data, the open response item, and the closed-ended questions to provide suggestive evidence of mechanisms through which the discussions reached their depolarizing effect. I use the chats data to see whether respondents ended up agreeing, finding common ground, or disagreeing. This additional analysis was not pre-registered. The open-response item and closed-ended questions provide insight into discussion experiences, that may correlate with treatment effects. These measures are taken during or after the discussion and cannot be considered unambiguous causal factors. At the same time, the

wait-list control design does avoid post-treatment bias, because the control group also gets treated. Thus, the design allows for a comparison of respondents in the control group who, if treated, would—for instance—attempt to find common ground, to respondents in the treatment group who do attempt to find common ground. This analysis requires the assumption that untreated respondents who end up finding common ground in the control group, would also have found common ground had they been assigned to the treatment group. Supplementary Table 23 shows that this assumption holds: treatment assignment is uncorrelated with scores on any of the mechanism questions. Thus, in this section, I estimate Conditional Average Treatment Effects (CATE): treatment effects conditional on the presence or absence of certain mechanisms. For outcome measures, I focus on the sympathy and discussion variables.

*Expressed disagreement*. A clear mechanism that appears from the analysis thus far, is issue agreement. Treatment effects on sympathy have been shown to depend on issue distance between discussing partisans. Here, I look at expressed disagreement in the discussion. To measure disagreement in the discussion, I conduct an inductive qualitative analysis of each chat discussion and see whether discussing partisans attempt to find common ground, or air differences. I create three categories: agreement, common ground, and disagreement. In the agreement category, (49% of discussions) participants express only agreement with each other's messages. Participants are put in the common ground category (27%) when they express disagreement, but come to understand each other's views, or steer the discussion towards points of consensus. In the disagreement category, (24%), respondents solely express opposing views. Expressed disagreement of course correlates with issue disagreement, but the two do not fully overlap. In 39% of cases in which participants' opinions are similar, the discussion expresses some disagreement; in 34% of cases in which participants' opinions are different, they only express points of agreement in the discussion. Supplementary Table 24 shows that whether respondents disagree or find common ground is only very slightly predicted by issue distance, and education levels. The Table additionally shows that covariates between treatment and control are balanced within each disagreement category.

Figure 6 shows the effect of the discussion on sympathy and willingness to discuss, by expressed disagreement. Results show that the treatment only increases sympathy when partisans agree or find common ground (t(290) = −4.11, P < 0.001, CATE = −1.68, 95% CI = (−2.49−−0.87)). When they disagree, treatment and control groups do not significantly differ and the coefficient for the treatment group is even slightly below that of the control group. On the discussion outcome, there is no statistically significant difference in treatment effects by disagreement. Supplementary Table 25 shows concomitant regression models.

*Additional mechanisms*. To measure additional hypothesized mechanisms, I use closed-ended questions asked after the discussion. They ask respondents about their perception of the behavior of their discussion partner in the discussion (e.g. Inclusion: 'my discussion partner wanted to hear what I had to say') or their perception of certain aspects of the discussion (e.g. Perspective-getting: 'The discussion made me see things from the perspective of my discussion partner'). These questions tap into the presence of perspective-getting, inclusion in the conversation, and commonality. I test their relation to the depolarizing effect of these discussions. The closed-ended questions were preceded by an open-ended question that asked respondents about their discussion experience. Respondents generally gave elaborate answers, and wrote an average of 80 words. One mechanism that

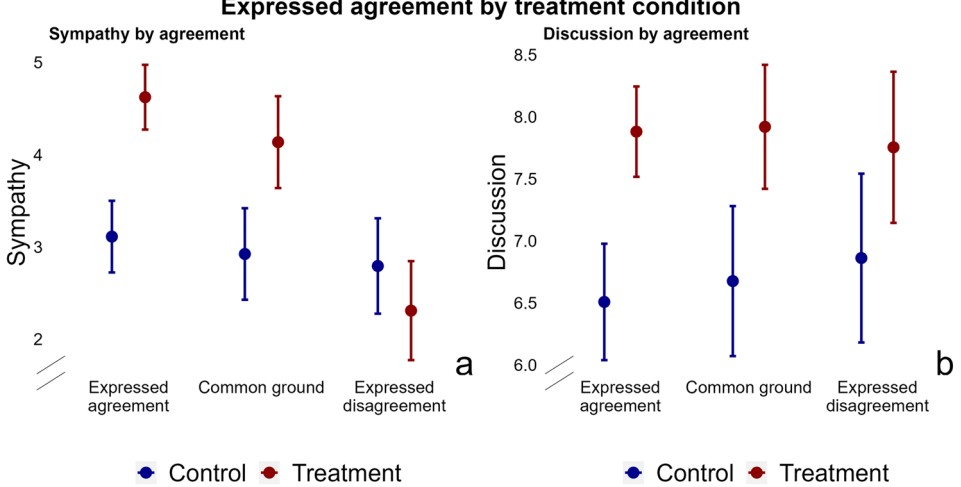

**Fig. 6 Expressed disagreement is associated with a significantly smaller effect of the discussion on out-partisan sympathy, but not on willingness to discuss.** Point estimates and 95% CIs are shown for the control group (blue) and the treatment group (red). Models interact the treatment with a variable capturing expressed disagreement. The variable has three values: agreement, common ground and disagreement. I report interaction terms, as I hypothesize that treatment effects vary by disagreement condition. The model controls for baseline covariates (gender, education, ethnicity, political interest, outparty closeness, partisan strength), and clusters SEs at the dyad level. Compared to respondents who agree, the discussion does not increase out-partisan sympathy for respondents who disagree ((**a**) (t(290) = −4.11, P < 0.001, CATE = −1.68, 95% CI = (−2.49-−0.87))). There is no statistically significant difference in treatment effects between those who find common ground and those who agree ((**a**) (t(290) = −0.83, P = 0.41, CATE = −0.34, 95% CI = (−1.16- 0.47))). On the discussion outcome, there is no statistically significant difference in treatment effects by disagreement (Common ground: ((**b**) (t(290) = −0.44, P = 0.66, CATE = −0.21, 95% CI = (−1.17-0.75)))); Disagreement; ((**b**) (t(290) = −1.42, P = 0.16, CATE = −0.7, 95% CI = (−1.7-0.28))).

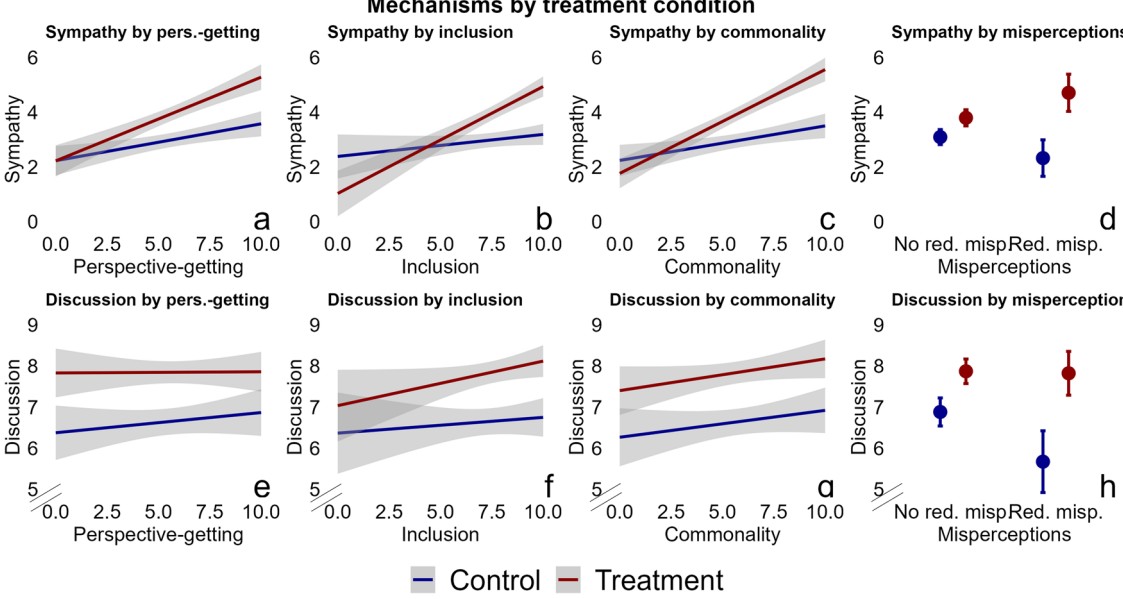

**Fig. 7 Experiencing perspective-getting, political inclusion, commonality and reduced misperceptions is associated with a larger effect of the discussion on out-partisan sympathy, but not on willingness to discuss.** Regression lines and 95% CIs (shading) are shown for the control group (blue) and the treatment group (red). Models interact treatment effects with each post-treatment mechanism question. I report interaction terms because I test whether treatment effects depend on the presence or absence of these mechanisms. The models control for baseline covariates (gender, education, ethnicity, political interest, outparty closeness, partisan strength), and cluster SEs at the dyad level. Results show that the treatment effect on out-partisan sympathy is significantly related to experiences of perspective-getting ((**a**) (t(296) = 2.73, P = 0.007, CATE = 0.19, 95% CI = (0.05-0.3))), feeling included in the conversation ((**b**) (t(296) = 4.36, P < 0.001, CATE = 0.28, 95% CI = (0.16-0.41))), feelings of commonality with one's discussion partner ((**c**) (t(296) = 4.27, P < 0.001, CATE = 0.23, 95% CI = (0.12-0.33))) and reduced misperceptions ((**d**) (t(296) = 4.03, P < 0.001, CATE = 1.88, 95% CI = (0.96-2.81))). There is no statistically significant difference in treatment effects on willingness to discuss by levels of either mechanism ((**e**) perspective-getting (t(296) = −0.6, P = 0.54, CATE = −0.04, 95% CI = (−0.18-0.09))), ((**f**) inclusion (t(296) = 1.3, P = 0.2, CATE = 0.1, 95% CI = (−0.06-0.27))), ((**g**) commonality (t(296) = 0.99, P = 0.32, CATE = 0.07, 95% CI = (−0.07-0.2))), with the exception of the misperceptions variable ((**h**) (t(296) = 2.03, P = 0.045, CATE = 1.09, 95% CI = (0.02-2.15))).

**Table 1 Sample Characteristics**

|  | Final sample (N = 582) | Pre-screener (N = 1697) | BES 23 (N = 16.217) |
|---|---|---|---|
| Women | 57% | 58% | 54% |
| Age (mean, sd) | 44,14 | NA | 55, 17 |
| Master/PhD | 15.9% | 16.1% | 10% |
| Bachelor | 39.9% | 37.3% | 36.7% |
| 5 or more good GCSEs | 27.5% | 27.5% | 34% |
| Other Education | 16.7% | 18.9% | 19.3% |
| Strong partisan | 32.1% | 31.6% | 15% |
| Moderate partisan | 52% | 51.1% | 51% |
| Weak partisan | 15.6% | 17.1% | 34% |
| Very interested in politics | 41.4% | 35.1% | NA |
| Moderately interested in politics | 53.3% | 59.3% | NA |
| Not interested in politics | 5% | 5.3% | NA |
| Ideology Labour | 3.4, 1.6 | 3.4, 1.6 | 3.3, 1.8 |
| Ideology Conservative | 6.5, 1.3 | 6.5, 1.3 | 6.7, 1.5 |
| Baseline out-party affect | 2.8, 2.2 | NA | 1.6, 1.9 |
| Immigration Labour | 3.5, 1.1 (1–5) | 3.4, 1.1 | 5.8, 2.7 (1–10) |
| Immigration Conservative | 2, 1 | 2, 1 | 2.7, 2.5 |
| Redistribution Labour | 4, 1.1 | 4, 1 | 4, 1 (1–5) |
| Redistribution Conservative | 2.5, 1.2 | 2.5, 1.2 | 2.6, 1.1 |

I did not test with a closed-ended statement is clearly visible in the open-response item. About 18% of respondents indicated that the discussions reduced their misperceptions about the policy views of out-partisans and the civility of the discussion. They wrote that they were apprehensive at first, expecting an extreme partner and a divisive discussion, but were pleasantly surprised and put at ease as soon as the discussion got underway. To test the relationship between these mechanisms and the depolarizing effect of the discussion, I again leverage the wait-list design to estimate treatment effects (CATE) conditional on mechanisms.

Figure 7 shows the relationship between mechanisms and both outcomes, for both treatment conditions. The slopes of the control group show that those with higher baseline sympathy were more likely to take a positive view of their partner's discussion behavior after the discussion. The slopes of the treatment group increase more steeply, and the difference in slopes shows the CATE. The treatment effect on out-partisan sympathy is clearly related to perceived perspective-getting (t(296) = 2.73, P = 0.007, CATE = 0.19, 95% CI = (0.05–0.3)), feeling included in the conversation (t(296) = 4.36, P < 0.001, CATE = 0.28, 95% CI = (0.16–0.41)), feelings of commonality (t(296) = 4.27, P < 0.001, CATE = 0.23, 95% CI = (0.12–0.33)) and reduced misperceptions (t(296) = 4.03, P < 0.001, CATE = 1.88, 95% CI = (0.96-2.81)). The same is not found for the willingness to discuss outcome, with the exception of reduced misperceptions (t(296) = 2.03, P = 0.045, CATE = 1.09, 95% CI = (0.02–2.15)). Supplementary Tables 26 and 27 show the concomitant regression models.

Finally, respondents who report reduced misperceptions were much more likely to have agreed, both in terms of pre-discussion issue distance, and during the discussion itself. This lends further credence to the idea that agreement, by reducing misperceptions, is an important driver of the effect of the discussion on out-partisan sympathy. Only 7% of respondents who report reduced misperceptions, ended up disagreeing in the discussion. 27% of respondents who did not report reduced misperceptions ended up disagreeing.

## Discussion
This study has shown that respondents who engage in a 10-minute, largely unmoderated, and potentially disagreeable cross-partisan discussion, show reduced levels of political polarization compared to a waitlist control group. After the discussion, respondents reported higher levels of out-partisan sympathy, and increased willingness to discuss across party lines. The discussion did not significantly affect issue opinions. These results add to the literature on inter-group contact and deliberation, by showing that interactions need not always be sustained or lengthy for them to reduce prejudice or polarization. Under certain conditions, even brief, purposeful interactions can have salutary effects. In addition, most respondents naturally—that is, without being explicitly asked to do so—engage in a range of conciliatory behaviors during the discussion. They often attempt to find common ground, share their perspectives and include each other in the conversation.

Most effects of the discussion are short-lived, pointing to a previously mentioned tension between scalability and durability of interventions[13]. At the same time, respondents do remain significantly more willing to have out-partisan discussions 2–3 weeks after the experiment. This points to the potential of a positive feedback loop, in which people are initially invited to test the waters in a brief exercise, and a positive experience makes them return for a longer discussion. Future work can study how to best convince out-partisans to engage with each other in the first place.

The effect of the discussion on out-partisan sympathy is contingent on issue distance between discussing partisans. Indeed, issue agreement may be an important mechanism driving the salutary effect of the discussion found in this study and in previous studies. This finding echoes studies which show that competitive or negative contact does not reduce prejudice[40,65], as well as a wider concern that contact interventions do not work on those most prejudiced[66]. At the same time, I find no statistically significant evidence of a backlash[21,40]. Furthermore, issue disagreement does not fully negate the depolarizing effect of the discussion. The discussion did increase willingness to have out-partisan discussions for all respondents, irrespective of issue disagreement. More generally, absent moderation, three out of four discussions were civil. Many who disagreed substantively tried to find common ground. Overall, results suggest that partisans can be encouraged to discuss across differences in unmoderated settings, and more research is needed on when and why discussions in which partisans disagree, become disagreeable.

A causal test of mechanisms was not incorporated into the design, but an exploration of rich qualitative data suggest that participants engaged in a range of conciliatory behaviors that

were associated with depolarization. In line with literature on misperceptions and partisan stereotypes, I find larger treatment effects on out-partisan sympathy for participants who reported being pleasantly surprised that they agreed on the issues, or that the discussion was civil. Similarly, perspective-getting, inclusion and feelings of commonality were all widely reported, and were all associated with increased out-partisan sympathy.

The finding that Conservative voters moderated their views in anticipation of the discussion deserves more attention. In real life, though people prefer to avoid political disagreement[62,67], they cannot always do so[63,68]. One strategy that people use when disagreement may arise, is to not disclose their political views. This behavior has been demonstrated in close social networks, in an effort to maintain good relationships[17,69], and in experimental settings, where people can observe a group norm and adapt to it[70,71]. This setting differs in two ways. First, discussants have no relationship to each other, nor is there a clearly observable group norm. Second, people did not have the option to not disclose their views: they could not start the discussion without first reporting their opinion. The results suggest yet another strategy of avoiding conflict, by disclosing a more moderate version of one's view going into, or during, the discussion. It is interesting that this behavior is found even in a brief, anonymous setting, suggesting, at least for some people, a deep-rooted desire to avoid a contentious discussion. This may in part explain why it was especially Conservative voters who moderated their views, as especially conservative immigration attitudes are somewhat stigmatized.

The study provides an example of an intervention to partisan animosity that is reasonably effective, and has already been implemented in society at large. A brief discussion at least momentarily lessens hostility for most participants, and the positive experience can be used to encourage more structured participation. While the effect of one brief discussion may soon whither, several positive experiences may more durably foster positive feelings towards the other side.

**Limitations**. Future work can improve upon this study in several ways. Methodologically, despite having several advantages, the wait-list control design hampers a proper measurement of persistence of effects because the control group also gets treated. In this case, it is possible that the UK political turmoil of June 2022 influenced polarization measures, though it is hard to explain why it would only affect willingness to discuss, and no other outcomes. Related, opinions in the control group were measured a few days before those in the treatment group. In addition, a slightly larger number of observations is required to make more precise estimates for certain sub-group analyses, most notably for the analysis of the effect of the treatment by issue distance and baseline opinion strength. In terms of generalizability, the UK is not as polarized as the US. At the same time, in this study, baseline animosity was high. In addition, two recent working papers test a chat-based intervention in more polarized contexts and find similar effects[72,73]. The study may also not generalize to those who are generally unwilling to communicate by chat with a stranger. It would finally be interesting to see whether a purposeful, light-touch intervention such as this one, also works in settings that involve group discussions or social identifiability. It could be that this setting lends itself especially well to a purposeful interaction because it largely frees participants from additionally having to navigate various social dynamics.

## Data availability

Anonymized replication is available at this link: https://osf.io/q4hjd/. Anonymized transcripts of the chat conversations are available upon request via email to the corresponding author.

## Code availability

The code to replicate the results is available this link: https://osf.io/q4hjd/.

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

## Acknowledgements

I am grateful to Arnout van de Rijt, Hanspeter Kriesi, Dylan Potts, Anna Clemente, Eelco Harteveld, Elias Dinas, Jamie Druckman, Matt Levendusky, Anica Waldendorf, Jonne Kamphorst, Natalia Garbiras-Díaz and the anonymous reviewers for comments and discussions. I also thank participants and discussants at EuroWEPS 2022, APSA 2022, EPSA 2022, ECPR 2021 Joint Sessions on affective polarization and NYU's Center for Social Media and Politics. I finally thank Leon Horbach, Ruben Treurniet and Jochem Tolenaar from Civinc for the collaboration. I am grateful to have received financial support from the European University Institute and the Dutch ministry of the Interior. Civinc and funders had no role in study design, data collection and analysis, decision to publish or preparation of the manuscript.

## Author contributions

J.F.d.J. designed research, performed research, analyzed data, wrote the paper, and provided revisions.

## Competing interests

The author declares no competing interests.
