## [Peer Review File · Communications Psychology]

9th Aug 23

Dear Mr de Jong,

Thank you for your patience during the peer-review process. Your manuscript titled "Brief, unmoderated cross-partisan discussions can reduce political polarization" has now been seen by 2 reviewers, and I include their comments at the end of this message. They find your work of interest, but raised some important points. We are interested in the possibility of publishing your study in *Communications Psychology*, but would like to consider your responses to these concerns and assess a revised manuscript before we make a final decision on publication.

We therefore invite you to revise and resubmit your manuscript, along with a point-by-point response to the reviewers. Please highlight all changes in the manuscript text file.

Editorially, we consider it important that you provide the additional analyses requested by the referees; please note that Reviewer #1 and #2 disagree regarding the appropriate analytical treatment of pre-intervention distance. It is up to you how you respond to these concerns (which suggestion for additional analysis you find scientifically more suitable), but please provide a clear rationale for your choice. We also ask you to address Reviewer #2's concerns regarding the persistence of effects and to tone down any inference on mechanism in the Discussion. Reviewer #1 asks for an overview table of past literature. While we agree that the literature review must be clear and comprehensive, we would not necessarily recommend such a table, as it could inadvertently give the appearance of a systematic review, which isn't the case. Reviewer #2 asks for parts of the analysis to be moved to the Supplement. Instead, we ask you to maintain all analyses in the main manuscript, but to discuss the limitations of each transparently in a section titled "Limitations" in the Discussion.

As you implement the revisions, please consult the checklist and template linked below to ensure the manuscript complies with all requirements. If the manuscript does not comply with our policies, especially with regard to statistics reporting and interpretation, and information on code and data availability, it may be returned to you.

With your resubmission, please provide a corrected Editorial Policy Checklist (the present checklist is missing a few entries).

The Reporting Summary needs to be updated to include information on Data Collection (currently empty).

Please use the following link to submit your revised manuscript, point-by-point response to the referees' comments (which should be in a separate document to any cover letter) and the completed checklist:

[Link redacted]

We hope to receive your revised paper within 8 weeks; please let us know if you aren't able to submit

it within this time so that we can discuss how best to proceed. If we don't hear from you, and the revision process takes significantly longer, we may close your file. In this event, we will still be happy to reconsider your paper at a later date, provided it still presents a significant contribution to the literature at that stage.

Please do not hesitate to contact me if you have any questions or would like to discuss these revisions further. We look forward to seeing the revised manuscript and thank you for the opportunity to review your work.

Best regards,

Marike

Marike Schiffer, PhD
Chief Editor
Communications Psychology

EDITORIAL POLICIES AND FORMATTING

Editorial Policy: [Policy requirements](https://www.nature.com/documents/nr-editorial-policy-checklist.pdf) (Download the link to your computer as a PDF.)

Furthermore, please align your manuscript with our format requirements, which are summarized on the following checklist:

[Communications Psychology formatting checklist](https://www.nature.com/documents/commspsychol-style-formatting-checklist-article-rr.pdf)

and also in our style and formatting guide [Communications Psychology formatting guide](https://www.nature.com/documents/commspsychol-style-formatting-guide-accept.pdf) .

* **CODE AVAILABILITY:** All Communications Psychology manuscripts must include a section titled "Code Availability" at the end of the methods section. In the event of publication, we require that the custom analysis code supporting your conclusions is made available in a publicly accessible repository;

at publication, we ask you to choose a repository that provides a DOI for the code; the link to the repository and the DOI will need to be included in the Code Availability statement. Publication as Supplementary Information will not suffice. We ask you to prepare code at this stage, to avoid delays later on in the process.

*** DATA AVAILABILITY:**

All Communications Psychology manuscripts must include a section titled "Data Availability" at the end of the Methods section or main text (if no Methods). More information on this policy, is available at <http://www.nature.com/authors/policies/data/data-availability-statements-data-citations.pdf>.

At a minimum the Data availability statement must explain how the data can be obtained and whether there are any restrictions on data sharing. Communications Psychology strongly endorses open sharing of data. If you do make your data openly available, please include in the statement:

We recommend submitting the data to discipline-specific, community-recognized repositories, where possible and a list of recommended repositories is provided at <http://www.nature.com/sdata/policies/repositories>.

If a community resource is unavailable, data can be submitted to generalist repositories such as [figshare](https://figshare.com/) or [Dryad Digital Repository](http://datadryad.org/). Please provide a unique identifier for the data (for example a DOI or a permanent URL) in the data availability statement, if possible. If the repository does not provide identifiers, we encourage authors to supply the search terms that will return the data. For data that have been obtained from publicly available sources, please provide a URL and the specific data product name in the data availability statement. Data with a DOI should be further cited in the methods reference section.

REVIEWERS' EXPERTISE:

Both reviewers work in the domain of polarization, in-group/out-group contact, and related topics

REVIEWERS' COMMENTS:

Reviewer #1 (Remarks to the Author):

This is a very nice paper which presents the results of an evaluation of a discussion intervention in the UK. Although there are now several papers examining the effects of such interventions which the author cites, what stands out about this paper is that the intervention was naturally occurring instead of having been conducted by researchers, and that the instructions to participants were different than

used in prior research. The findings are promising, contrasting with prior research that has found political conversations are less effective. I think this is a very strong contribution to the emerging literature on cross-partisan discussions.

I support the publication of this paper with minor revisions as follows:

1. Participants were not given no prompt, they were told to discuss one of two issues. It seems to me that a very plausible explanation for the results is that the participants actually agreed with each other on the issues, and then learned that they agreed more than they thought. It would be nice to see a test of this hypothesis by looking at the effects by the distance between their initial positions on the items. One reason some previous authors in this literature haven't used prompts about specific issues is the concern that the effects would just be driven by agreeing on the issue, which might not generalize to contexts where two groups are defined by disagreement on an issue. Either way, it would be good to know what is found here.

2. The paper contrasts itself with the results of the previous literature, but I think a reader who isn't deeply immersed in this literature would have a hard time understanding what exactly is different between this paper and previous work. The author should make a lit review table that has each condition in a previous study on one row, its findings in one column, and then one column each for the various important dimensions on which this study and previous studies differ -- the prompts, lengths, samples, mode of interaction, etc.

Smaller comments:

3. I found that there was too much emphasis on scalability in the manuscript. That is an important issue, no doubt, but there is very little analysis in the manuscript of what made this scalable, how scalable it truly was, etc. I think this can be mentioned, but to make it the main focus when the empirics are not about that seems misplaced.

4. There should be an acknowledgement that the effects might not generalize to those who do not volunteer to participate in prejudice reduction interventions.

5. A very nice feature of the design is that the author can do both between- and within-subject comparisons; the former being better for judging impact, the latter being necessary for looking at what features of the conversations most correlated with change. The figure captions should be explicit as to whether between- or within-subject variation is being examined in that figure.

6. Given the potential for regression to the mean to drive the findings of the within-subject analyses, it would be nice to see robustness checks on the within-subject analyses within a regression framework (i.e., $\text{post} \sim \text{pre} + \text{covariate}$, instead of using $\text{post} - \text{pre}$ as the outcome).

I would like to review the revision of the paper, but close by restating my support for the paper's eventual publication.

Reviewer #2 (Remarks to the Author):

Referee Report for "Brief, unmoderated cross-partisan discussions can reduce political polarization"

This study reports the result of an experiment that randomly exposed dyads consisting of one Labour voter and one Conservative voter to have unmoderated discussions about a contentious political issue (immigration or redistribution). The headline result is that the intervention has a large, 1 point (11 point scale) effect on outparty sympathy that is 0 points 2-3 weeks later. The intervention increases willingness to discuss politics by one point that is 0.5 points 2-3 weeks later. The intervention also moderated views, with conservatives expressing slightly more progressive views and progressives expressing slightly more conservative views.

I found the reporting to be very clear and the substantive importance of the problem the intervention is designed to solve is self-evident. I think the design is strong and can indeed reveal the causal effect of these conversations. I also found it fascinating that just telling respondents who they were going to have a conversation with caused a change in expressed opinion.

I have comments to offer in four areas:

1. On persistence.

In the discussion section, the author writes: "The NGO I collaborated with preferred a wait list control design, which ensures that all respondents take the treatment, but hampers a proper measurement of persistence of effects because the control group also gets treated." I think this design flaw is serious - in what we are measuring the overtime effect of treatment if we can't rely on a treatment versus control difference. It seems like the 2-3 weeks rows in figure 3 compared treated subjects to treated-10-minutes-later subjects, so I don't think we learn about the persistence of the effect of the treatment on attitudes. Unless I'm misunderstanding, I think the claims in the abstract and in the main paper about persistence are not supported by the design.

2. Within / between subjects confusion.

The design allows for between-subjects comparisons (treatment group to control group) and within-subjects comparisons (in the control group, the post-pre change.) Please only present the between-subjects results in the main paper. The comparison to the within-subjects design is of some interest, but in my view, the main paper should focus on the inferences we can draw on the basis of the random assignment.

3. Change scores

Even in the between-subjects design, the analysis operates on post - pre difference in attitudes (the change score). At times, the manuscript veers into describing these change scores as estimating treatment effects (Figure 8, for example, is confusing on this point because the vertical axis is the change score.)

In my view, the pre-treatment level of the attitude is just a covariate like any other, and we could adjust our treatment effect estimates using that covariate. So you could have the difference in means estimate of the effect from a regression of Y_{post} on treatment or the covariate adjusted estimate from a regression of Y_{post} on treatment plus Y_{pre} (One could also use the Lin (2013) version of this estimator). Running the regression of $(Y_{post} - Y_{pre})$ on treatment is of course unbiased for the ATE, but it has lower precision than the covariate adjusted estimator. Not using change scores as the outcome *clarifies* that the random assignment is the basis for inference, not the before-after comparison within the treatment group.

4. Discussion features

The "Disagreement" analysis beginning on page 10 and the "Mechanisms" analyses beginning on page 13 are conditioned on features of the conversation that developed as the dyads interacted. Conditioning on post-treatment variables in this way usually causes bias. If the design works the way I think it does, this study actually avoids this post treatment bias problem (ironically enough!) because of the wait list design. We can compare dyads who agree in the treatment to dyads who agree in the control group, only because the control group also (eventually) experiences treatment.

I would encourage the author to describe the estimates in this section as "conditional average treatment effect estimates" because they apply to the subjects of dyads who (if treated) would agree, would engage in perspective getting, etc.

However, this analysis requires an assumption that a treated dyad who expresses agreement would *also* have expressed agreement had they been assigned to the control group. Please formally state this assumption in the paper. This assumption could be violated (for example) if the act of answering the outcome survey changes the quality of the conversations. Luckily, the design can assess that directly. Please demonstrate that treatment does not increase or decrease the prevalence of these discussion features by comparing their rates in the treatment and control groups. Further, please demonstrate covariate balance within each of the "strata" formed by discussion features. If the treatment and control group dyads who agree appear to be different from one another, then the disagreement and mechanisms analyses *are* subject to post-treatment bias concerns.

I would also encourage the author to be circumspect about what we can learn about mechanisms from these analyses. The kinds of dyads who end up agreeing or end up disagreeing may be systematically different from one another, so it's possible that the differences in effects on the outcome by agreement / disagreement are reflections of *who* the dyads are, not the "mechanism" of agreement per se.

REVISION MEMO

'BRIEF, UNMODERATED CROSS-PARTISAN DISCUSSIONS CAN REDUCE POLITICAL POLARIZATION'

Nature Communications Psychology

Introduction

Dear Reviewers,

Thank you for your thorough and thoughtful comments and suggestions, which helped to significantly improve the manuscript. Below, I explain how I addressed each of the points raised. Before doing so, I want to highlight two larger changes to the manuscript.

First, I added a more elaborate section on the effects of the intervention by baseline issue distance between opposing partisans. The results are in a new Figure 4 and show that treatment effects on out-partisan sympathy depend on baseline issue distance between discussing partisans.

Second, I changed the models and identification strategy for the results on opinion change. I no longer use a within-design, nor within-individual change scores as the outcome measure. Instead, I compare baseline opinions in the control group to post-treatment opinions in the treatment group. I no longer find evidence for opinion change when using this method.

Comments by Reviewer 1

- 1. Participants were not given no prompt, they were told to discuss one of two issues. It seems to me that a very plausible explanation for the results is that the participants actually agreed with each other on the issues, and then learned that they agreed more than they thought. It would be nice to see a test of this hypothesis by looking at the effects by the distance between their initial positions on the items. One reason some previous authors in this literature haven't used prompts about specific issues is the concern that the effects would just be driven by agreeing on the issue, which might not generalize to contexts where two groups are defined by disagreement on an issue. Either way, it would be good to know what is found here.*

- **Reply:** I have taken out the sentence that participants were given no prompt. I agree that this explanation for the results is plausible, given the US literature on how correcting misperceptions about ideology, social composition, and policy, reduces affective polarization [1–3]. I added a section in the manuscript devoted to issue disagreement called 'discussion effects by issue disagreement'. Figure 4 shows effects by agreement, and regression models are in Supplementary Tables 6 and 7. Overall, the effect of the treatment on out-partisan sympathy indeed depends on issue distance between discussing partisans. More in particular, there is a pronounced difference in effects size among respondents with strong baseline opinions, depending on whether they were randomly matched to someone who agreed or disagreed with them. For those with moderate opinions, or for other outcomes, I find no evidence that disagreement affects treatment effects. I also point to my response to the fourth comment by Reviewer 2.

- 2. The paper contrasts itself with the results of the previous literature, but I think a reader who isn't deeply immersed in this literature would have a hard time understanding what exactly is different between this paper and previous work. The author should make a lit review table that has each condition in a previous study on one row, its findings in one column, and then one column each for the various*

important dimensions on which this study and previous studies differ – the prompts, lengths, samples, mode of interaction, etc.

- **Reply:** I agree that I should be more clear on the difference in design and results between this study and previous studies. However, the editor does not recommend this specific table, because it would give the appearance of a systematic review. Instead, I have added more specific information on both the difference between this paper and previous work, and the difference in results in the introduction, in the final paragraph of the section titled 'The experiment', and in the discussion.

3. *I found that there was too much emphasis on scalability in the manuscript. That is an important issue, no doubt, but there is very little analysis in the manuscript of what made this scalable, how scalable it truly was, etc. I think this can be mentioned, but to make it the main focus when the empirics are not about that seems misplaced.*

- **Reply:** I agree that the main focus of the paper should not be about scalability. Mentioning scalability in the abstract and having it feature prominently in the introduction indeed draws too much attention to it. I have removed the sentence 'I show that this type of intervention is scalable' from the abstract, and I moved Figure 1 from the introduction to the beginning of the results section.

4. *There should be an acknowledgement that the effects might not generalize to those who do not volunteer to participate in prejudice reduction interventions.*

- **Reply:** I have put a version of this acknowledgement in the discussion section, under 'limitations'. However, I should note that participants did not know that they were to engage in a prejudice-reduction intervention. The recruitment text read: 'For this study, we want to ask you a number of questions again, and ask you to have a brief, anonymous chat conversation on a novel chat platform. Would you be willing to participate' (see Supplementary Section B4 for all survey questions). So, up until seconds before the discussion, participants did not know they were going to talk politics, talk to an out-partisan, or have a discussion instead of a conversation. I added in the manuscript that effects may not generalize to those who are unwilling to have chat conversations.

5. *A very nice feature of the design is that the author can do both between- and within-subject comparisons; the former being better for judging impact, the latter being necessary for looking at what features of the conversations most correlated with change. The figure captions should be explicit as to whether between- or within-subject variation is being examined in that figure.*

- **Reply:** Also following the second comment by Reviewer 2, I now only use the between-design in the main paper to estimate treatment effects.

6. *Given the potential for regression to the mean to drive the findings of the within-subject analyses, it would be nice to see robustness checks on the within-subject analyses within a regression framework (i.e., post - pre + covariate, instead of using post - pre as the outcome).*

- **Reply:** I agree that conditioning on baseline opinions in the within-analysis led to observed effects that could have been driven by regression to the mean. To mitigate this, I now use a between-analysis, and no longer use change scores as the outcome measure (see also the response to Reviewer 2's third comment).

7. *I would like to review the revision of the paper, but close by restating my support for the paper's eventual publication.*

- **Reply:** I thank the reviewer for the support, and for the very helpful comments.

Comments by Reviewer 2

1. *On persistence. In the discussion section, the author writes: "The NGO I collaborated with preferred a wait list control design, which ensures that all respondents take the treatment, but hampers a proper measurement of persistence of effects because the control group also gets treated." I think this design flaw is serious – in what what are we measuring the overtime effect of treatment if we can't rely on a treatment versus control difference. It seems like the 2-3 weeks rows in figure 3 compared treated subjects to treated-10-minutes-later subjects, so I don't think we learn about the persistence of the effect of the treatment on attitudes. Unless I'm misunderstanding, I think the claims in the abstract and in the main paper about persistence are not supported by the design.*

- **Reply:** I partly agree with this point, but think the design flaw is less serious. Measurement of persistence does rely on a treatment versus control difference, but measurements of both groups are taken at different points in time. I measure persistence by comparing those in the treatment condition 2-3 weeks after taking the treatment, to those in the control condition *before* taking the treatment. I fully agree that the ideal design would additionally include a measurement of outcome variables in an untreated control group, 2-3 weeks after the treated get treated. Compared to this design, my design requires the additional assumption that observed effects are not influenced by any other development in the 2-3 weeks since the treated take the treatment. I do not think this assumption is unreasonable to make. No other outcome variable measured in this way is still significant 2-3 weeks later. It is hard to imagine a collective event that increased willingness to engage with out-partisans, but did not affect any of the other, related variables. Note also that the effect is there regardless of using a between, or within-design. For now, I have elaborated on this assumption in the discussion section, under 'limitations'. I have additionally softened claims about persistence in the introduction, interpretation of results, and discussion, stating that these results require an additional assumption and should be interpreted with caution. If the reviewer is unconvinced of my argumentation and remains of the opinion that the claims in the abstract and paper about persistence should be taken out, I can take them out.

2. *Within / between subjects confusion. The design allows for between-subjects comparisons (treatment group to control group) and within-subjects comparisons (in the control group, the post-pre change.) Please only present the between subjects results in the main paper. The comparison to the within-subjects design is of some interest, but in my view, the main paper should focus on the inferences we can draw on the basis of the random assignment.*

- **Reply:** The main paper now only presents results from the between-subjects design. All within-subjects results are presented in the appendix.

3. *Change scores. Even in the between-subjects design, the analysis operates on post - pre difference in attitudes (the change score). At times, the manuscript veers into describing these change scores as estimating treatment effects (Figure 8, for example, is confusing on this point because the vertical axis is the change score). In my view, the pre-treatment level of the attitude is just a covariate like any other, and we could adjust our treatment effect estimates using that covariate. So you could have the difference in means estimate of the effect from a regression of Y_{post} on treatment or the covariate adjusted estimate from a regression of Y_{post} on treatment plus Y_{pre} (One could also use the Lin (2013) version of this estimator). Running the regression of $(Y_{post} - Y_{pre})$ on treatment is of course unbiased for the ATE, but it has lower precision than the covariate adjusted estimator. Not using change scores as the outcome *clarifies* that the random assignment is the basis for inference, not the before-after comparison within the treatment group.*

- **Reply:** I agree with this point. I no longer use change scores, nor the within-design, to estimate treatment effects in any of the models. Though I also agree that a covariate adjusted estimator would be more precise, I did not collect baseline outcome measures for the treatment group, with the exception of issue opinions. Thus, I cannot estimate the covariate-adjusted estimator with my design.

4. *Discussion features. The "Disagreement" analysis beginning on page 10 and the "Mechanisms" analyses beginning on page 13 are conditioned on features of the conversation that developed as the dyads interacted. Conditioning on post-treatment variables in this way usually causes bias. If the design works the way I think it does, this study actually avoids this post treatment bias problem (ironically enough!) because of the wait list design. We can compare dyads who agree in the treatment to dyads who agree in the control group, only because the control group also (eventually) experiences treatment. I would encourage the author to describe the estimates in this section as "conditional average treatment effect estimates" because they apply to the subjects of dyads who (if treated) would agree, would engage in perspective getting, etc. However, this analysis requires an assumption that a treated dyad who expresses agreement would *also* have expressed agreement had they been assigned to the control group. Please formally state this assumption in the paper. This assumption could be violated (for example) if the act of answering the outcome survey changes the quality of the conversations. Luckily, the design can assess that directly. Please demonstrate that treatment does not increase or decrease the prevalence of these discussion features by comparing their rates in the treatment and control groups. Further, please demonstrate covariate balance within each of the "strata" formed by discussion features. If the treatment and control group dyads who agree appear to be different from one another, then the disagreement and mechanisms analyses *are* subject to post-treatment bias concerns. I would also encourage the author to be circumspect about what we can learn about mechanisms from these analyses. The kinds of dyads who end up agreeing or end up disagreeing may be systematically different from one another, so it's possible that the differences in effects on the outcome by agreement / disagreement are reflections of *who* the dyads are, not the "mechanism" of agreement per se.*

- **Reply:** These points are very well taken. The CATE design is an excellent idea. Figure 6 in the main paper actually already used this design for the expressed disagreement variable, but I did not make this sufficiently clear. I have provided more clarity in the preceding text, and formally stated the assumption the reviewer suggests. I test this assumption in Supplementary Tables 20 and 21. The first table shows that disagreement, and responses to the mechanisms questions are uncorrelated with treatment assignment. The second shows that each stratum of the expected disagreement variable is balanced across experimental conditions, that is: not predicted by any covariates. I have also called the results in figures 6 and 7 conditional average treatment effect estimates.
- Figures 8 and 9 in the previous version, which are now combined into Fig. 7, did not yet use the design the reviewer proposes. I changed the models to be in line with this design. The models now no longer use the change score as the outcome, but regress the same outcome variables used in all models, on the treatment and an interaction term for each mechanism. Results change slightly: the mechanisms are only related to the sympathy outcome, not to the discussion outcome.
- I agree that these mechanisms cannot be considered a causal factor, as they were not fully incorporated in my randomization design. I have changed relevant sections of the introduction, interpretation of results and discussion to not make causal inferences on mechanisms.
- The exception here is baseline opinion distance between discussing partisans. This measure is taken before the treatment, and though it naturally depends on a respondent's initial position, their partner is randomly assigned, and thus so is their opinion. For instance, someone who

strongly disagrees with a statement can be matched to someone who is neutral, or to someone who strongly agrees. In contrast to the other models, then, the models that look at treatment effects by opinion distance can be interpreted causally, as long as I control for a respondent's baseline opinions. I elaborate on this point in the section now titled 'discussion effects by issue disagreement'.

- I want to thank the reviewer very much for their comments.

References

- [1] Douglas J Ahler and Gaurav Sood. "The parties in our heads: Misperceptions about party composition and their consequences". In: *The Journal of Politics* 80.3 (2018), pp. 964–981.
- [2] Lilla V Orr and Gregory A Huber. "The policy basis of measured partisan animosity in the United States". In: *American Journal of Political Science* 64.3 (2020), pp. 569–586.
- [3] James N Druckman et al. "(Mis) estimating affective polarization". In: *The Journal of Politics* 84.2 (2022), pp. 1106–1117.

Decision letter and referee reports: second round

12th Oct 23

Dear Mr de Jong,

Thank you for your patience during the peer-review process. Your manuscript titled "Brief, unmoderated cross-partisan discussions can reduce political polarization" has now been seen by the same 2 reviewers as before, who very kindly rapidly provided us with an evaluation of your revisions. I include their comments at the end of this message.

The referees find your work much improved, but point to some presentational issues and a few methodological queries. We are interested in the possibility of publishing your study in *Communications Psychology*. Before we make a final decision, we ask you to revise the manuscript one last time to address these issues and format the manuscript according to our guidelines (to which I link below).

We therefore invite you to revise and resubmit your manuscript, along with a point-by-point response to the reviewers. Please highlight all changes in the manuscript text file.

Reviewer #1 raises an important issue with regard to the presentation of existing work. As a general stylistic point, we recommend focusing on the features of the present study, while existing studies should be faithfully described to allow the reader to situate the work in its relevant literature. A potential discussion of perceived shortcomings of existing articles is often a superfluous distraction.

Please note that your revised manuscript must comply with our formatting and reporting requirements, which are summarized on the following checklist:

<https://www.nature.com/documents/commspsychol-style-formatting-checklist-article-rr.pdf> Communications Psychology formatting checklist and also in our style and formatting guide <https://www.nature.com/documents/commspsychol-style-formatting-guide-accept.pdf> Communications Psychology formatting guide .

Please use the following link to submit your revised manuscript, point-by-point response to the referees' comments (which should be in a separate document to any cover letter) and the completed checklist:

[Link redacted]

We would appreciate it if you could keep us informed about an estimated timescale for resubmission,

to facilitate our planning.

Please do not hesitate to contact me if you have any questions or would like to discuss these revisions further. We look forward to seeing the revised manuscript and thank you for the opportunity to review your work.

Best regards,

Marike

Marike Schiffer, PhD
Chief Editor
Communications Psychology

EDITORIAL POLICIES AND FORMATTING

Editorial Policy: [Policy requirements](https://www.nature.com/documents/nr-editorial-policy-checklist.pdf) (Download the link to your computer as a PDF.)

* **CODE AVAILABILITY:** All Communications Psychology manuscripts must include a section titled "Code Availability" at the end of the methods section. In the event of publication, we require that the custom analysis code supporting your conclusions is made available in a publicly accessible repository; at publication, we ask you to choose a repository that provides a DOI for the code; the link to the repository and the DOI will need to be included in the Code Availability statement. Publication as Supplementary Information will not suffice. We ask you to prepare code at this stage, to avoid delays later on in the process.

* **DATA AVAILABILITY:**

All Communications Psychology manuscripts must include a section titled "Data Availability" at the end of the Methods section or main text (if no Methods). More information on this policy, is available at <http://www.nature.com/authors/policies/data/data-availability-statements-data-citations.pdf>.

At a minimum the Data availability statement must explain how the data can be obtained and whether there are any restrictions on data sharing. Communications Psychology strongly endorses open sharing of data. If you do make your data openly available, please include in the statement:

We recommend submitting the data to discipline-specific, community-recognized repositories, where possible and a list of recommended repositories is provided at <http://www.nature.com/sdata/policies/repositories>.

If a community resource is unavailable, data can be submitted to generalist repositories such as [figshare](https://figshare.com/) or [Dryad Digital Repository](http://datadryad.org/). Please provide a unique identifier for the data (for example a DOI or a permanent URL) in the data availability statement, if possible. If the repository does not provide identifiers, we encourage authors to supply the search terms that will return the data. For data that have been obtained from publicly available sources, please provide a URL and the specific data product name in the data availability statement. Data with a DOI should be further cited in the methods reference section.

REVIEWERS' COMMENTS:

Reviewer #1 (Remarks to the Author):

I think the author did a fantastic job with the revisions and I continue to think this is a great paper. I only have three minor remaining issues:

1. The abstract claims that existing studies rely largely on "moderated, lengthy" interactions but I do not think this is true. E.g., the studies cited in 62, 65, 67 are all not moderated and not particularly lengthy. So I think the abstract needs to be changed to reflect the paragraph on p. 7 that highlights what is different about this study from other studies: 1) the study focuses on discussing a specific issue but allows the degree of disagreement to vary, and therefore allows us to see that issue agreement is important for facilitating these effects, and 2) looks at mechanisms using the within-subjects design. All the other features have been present in prior studies cited in the paper (lack of moderators, looking at long term effects, short discussions). I understand the Editor doesn't want a lit review table to appear in the paper, but the author should probably make one for themselves to look at even if it doesn't go in the paper, so they can be clear for themselves about what exactly is new and not here. There is definitely stuff that is new but what is new should be accurate.

2. The new results show that the effects on sympathy seem quite critically conditional on issue distance. However, the discussion of mechanisms focuses around conciliatory behavior. It seems like the obvious other mechanism here is greater perceptions of agreement with the other side on issues, given the heterogenous treatment effect results. Was perception of agreement with the other side measured? Even if not, but especially if so, this theme seems like it deserves more focus in the discussion of mechanisms. (Indeed, conciliatory behavior seems like another moderator, not a mechanism.)

3. The author says that the results for issue distance don't show up for those with moderate opinions, but we don't see the point estimate on the estimated interaction and standard error between treatment and issue distance for this subset to know if it is any different. In the figure note for Figure 4 we are told that it is insignificant, but the estimate is in the same direction. It is possible, therefore, that estimates for the interaction are actually quite similar for those with and without moderate issue opinions, there is just less statistical power among those with moderate opinions. This is very different than a finding that we can statistically distinguish between the effects of issue distance for those with and without moderate opinions. I.e., the author is arguing for a three-way interaction between treatment, issue distance, and issue moderation, but hasn't actually tested this, and this three-way interaction might be quite imprecisely estimated.

4. I don't understand why the control group is present in Figure 7. Isn't this looking at what predicts within-subject changes within the treatment group? Why is the relationship between the post-treatment moderator and the outcome in the control group relevant?

Reviewer #2 (Remarks to the Author):

Referee Report for "Brief, unmoderated cross-partisan discussions can reduce political polarization" (Nature Communications Psychology 282-1)

I thank the author for their thorough revision to their article. The only remaining point of possible disagreement was the persistence estimates. I had misunderstood the estimation approach in the original submission (imagining we were comparing T3 responses across treatment and control). I now understand that we are comparing T1 responses in the control group to T3 responses in the treatment group, just as we are comparing T1 responses in the control group to T2 responses in the treatment group for the immediate analysis. The required assumption (nothing else changes responses except treatment) is easier to believe when the time elapsed is 10 minutes and harder to believe when it's 2-3 weeks later. That said, I think the author does a very good job of explaining how to understand the persistence estimates and I don't think they should remove it just because it's not as good as comparing T3 in a never-treated control to T3 in the treatment.

Three quick comments:

1) for the persistence estimates do we compare the T3 responses in the treatment group to the T1 responses to those in the control group who responded to T3? or to the whole control group? I think it would be more appropriate to condition on responding at T3, to keep things as closely comparable as possible.

2) the bottom row of Figure 4 amounts to a triple interaction of treatment, issue distance, and opinion strength. While I see how the treatment x issue distance interaction seems strongest for those with the strongest opinions, I wonder about the uncertainty attending to the difference-in-difference-in-difference. The paragraph preceding the figure suggests that it works for those with strong opinions, but not the others, but I doubt this experiment has the precision to tell the difference. A formal hypothesis test of the triple interaction should accompany this claim one way or another.

3) I appreciate that we only have t1 pre-treatment responses in the control group and not the treatment group, but there's information in the t0 screener that presumably correlates somewhat with post-treatment outcomes; it wouldn't be crazy to squeeze out a little more precision by adjusting for those variables.

REVISION MEMO 2

'BRIEF, UNMODERATED CROSS-PARTISAN DISCUSSIONS CAN REDUCE POLITICAL POLARIZATION'

Nature Communications Psychology

Introduction

Dear Reviewers,

I want to thank you again for the very swift reply and the excellent comments. Below I address each of your points.

Comments by Reviewer 1

- 1. The abstract claims that existing studies rely largely on "moderated, lengthy" interactions but I do not think this is true. E.g., the studies cited in 62, 65, 67 are all not moderated and not particularly lengthy. So I think the abstract needs to be changed to reflect the paragraph on p. 7 that highlights what is different about this study from other studies: 1) the study focuses on discussing a specific issue but allows the degree of disagreement to vary, and therefore allows us to see that issue agreement is important for facilitating these effects, and 2) looks at mechanisms using the within-subjects design. All the other features have been present in prior studies cited in the paper (lack of moderators, looking at long term effects, short discussions). I understand the Editor doesn't want a lit review table to appear in the paper, but the author should probably make one for themselves to look at even if it doesn't go in the paper, so they can be clear for themselves about what exactly is new and not here. There is definitely stuff that is new but what is new should be accurate.*

- **Reply:** I have changed the abstract to be more in line with the paragraph formerly on p.7. I have done the same for the broader introduction, and smaller parts of the theory section, to which this point also applies. I have highlighted the two points you mention, and agree that these are the study's main contributions. I do think there are a few additional contributions that the study makes. First, it makes disagreement more visible than previous studies, by showing discussing partisans each other's views on the issues throughout the discussion. The Santoro and Broockman study of course encourages a more negative conversation by asking people to discuss partisanship, but that is different from issue disagreement. Second, it shows that some people moderate their views just before the discussion, which is I would argue an example of conciliatory behavior, a point to which I return below.

- 2. The new results show that the effects on sympathy seem quite critically conditional on issue distance. However, the discussion of mechanisms focuses around conciliatory behavior. It seems like the obvious other mechanism here is greater perceptions of agreement with the other side on issues, given the heterogenous treatment effect results. Was perception of agreement with the other side measured? Even if not, but especially if so, this theme seems like it deserves more focus in the discussion of mechanisms. (Indeed, conciliatory behavior seems like another moderator, not a mechanism.)*

- **Reply:** The discussion of mechanisms now contains more emphasis on the role of agreement. Perception of agreement was, unfortunately, not measured. A proxy could be the misperceptions measure, where participants reported, in an open response question asked after the treatment, that they were surprised to agree, and surprised about the civility of the discussion. Reporting

reduced misperceptions is correlated with treatment effects. In the final paragraph of the section titled 'mechanisms', I now show that those who report reduced misperceptions, are much more likely to have agreed in the discussion than those who do not report reduced misperceptions.

3. *The author says that the results for issue distance don't show up for those with moderate opinions, but we don't see the point estimate on the estimated interaction and standard error between treatment and issue distance for this subset to know if it is any different. In the figure note for Figure 4 we are told that it is insignificant, but the estimate is in the same direction. It is possible, therefore, that estimates for the interaction are actually quite similar for those with and without moderate issue opinions, there is just less statistical power among those with moderate opinions. This is very different than a finding that we can statistically distinguish between the effects of issue distance for those with and without moderate opinions. I.e., the author is arguing for a three-way interaction between treatment, issue distance, and issue moderation, but hasn't actually tested this, and this three-way interaction might be quite imprecisely estimated.*

- **Reply:** I now report results from a three-way interaction in the text above figure 4. It falls just below conventional standards for statistical significance ($p = 0.1$). There is thus indeed no statistically significant difference of the treatment effect on sympathy, by issue distance, between respondents with moderate and strong opinions.

4. *I don't understand why the control group is present in Figure 7. Isn't this looking at what predicts within-subject changes within the treatment group? Why is the relationship between the post-treatment moderator and the outcome in the control group relevant?*

- **Reply:** I no longer use within-subject changes in the control group to explore mechanisms. In the previous revision, Reviewer 2 pointed out that the wait-list design allows for an exploration of mechanisms while avoiding post-treatment bias. This is the case because the control group eventually gets treated, so I can compare those in the control group who, for example, *will* find common ground, to those in the treatment group who *do* find common ground. This hinges on the assumption that being on the wait-list does not change the probability of finding common ground. In Supplementary Table 23, I show that this assumption holds, by showing that treatment assignment does not affect presence or absence of these mechanisms. For a more elaborate treatment of this, I point to my response to Reviewer 2's fourth point in the previous response letter, and to the second paragraph of the section on mechanisms. In Figure 7 specifically, the lines for the control group show the correlation between a respondent's outcome at baseline, and their response to the mechanism question post-treatment. This is the counterfactual situation for the treatment group. The effect of the treatment by these mechanism can thus be observed by comparing the slopes of the blue and red lines.

Comments by Reviewer 2

1. *For the persistence estimates do we compare the T3 responses in the treatment group to the T1 responses to those in the control group who responded to T3? or to the whole control group? I think it would be more appropriate to condition on responding at T3, to keep things as closely comparable as possible.*

- **Reply:** I agree that this comparison is more appropriate and I changed the models to only include those in the control group who responded to the follow-up survey. The results do not change significantly.

2. *The bottom row of Figure 4 amounts to a triple interaction of treatment, issue distance, and opinion strength. While I see how the treatment x issue distance interaction seems strongest for those with the strongest opinions, I wonder about the uncertainty attending to the difference-in-difference-in-difference. The paragraph preceding the figure suggests that it works for those with strong opinions, but not the others, but I doubt this experiment has the precision to tell the difference. A formal hypothesis test of the triple interaction should accompany this claim one way or another.*

- **Reply:** The text above the model now shows results from this triple interaction model. The experiment indeed seems to lack precision to tell the difference: the P value of the triple interaction term is 0.1, and the confidence intervals are quite large. I have removed the statement on the differences between those with strong, and those with moderate opinions.

3. *I appreciate that we only have t1 pre-treatment responses in the control group and not the treatment group, but there's information in the t0 screener that presumably correlates somewhat with post-treatment outcomes; it wouldn't be crazy to squeeze out a little more precision by adjusting for those variables.*

- **Reply:** I added baseline feelings of closeness to the out-party, and party strength to all the models. Updated results often show a small increase in treatment effects for the sympathy-related variables. Some estimates that were borderline insignificant, are now significant.

21st Nov 23

Dear Mr de Jong,

Your manuscript titled "Brief, unmoderated cross-partisan discussions can reduce political polarization" has now been seen by our reviewers, whose comments appear below. In light of their advice I am delighted to say that we are happy, in principle, to publish a suitably revised version in Communications Psychology under the open access CC BY license (Creative Commons Attribution v4.0 International License).

We therefore invite you to revise your paper one last time to address the remaining editorial requests. At the same time we ask that you edit your manuscript to comply with our format requirements and to maximise the accessibility and therefore the impact of your work.

EDITORIAL REQUESTS:

- 1) Please revise your manuscript to ensure that strong causal language is avoided. More guidance on this is included in the "Editorial Requests Table".
- 2) Please deposit your Code and Data at this stage. More information on this is included in the Editorial Requests Table.
- 3) Please revise your reporting and interpretation of non-significant findings in null-hypothesis significance tests as explained in the Editorial Requests Table and on our statistics guidelines pages.
- 4) Please revise the presentation of the Methods and Results as follows. The Methods should precede the Results; both Methods and Results should be comprehensive, so that the reader can follow the presentation of evidence without referring to the SI.

Please review all specific editorial comments and requests regarding your manuscript in the attached Editorial Requests Table.

SUBMISSION INFORMATION:

OPEN ACCESS:

Communications Psychology is a fully open access journal. Articles are made freely accessible on publication under a [CC BY](http://creativecommons.org/licenses/by/4.0) license (Creative Commons Attribution 4.0 International License). This license allows maximum dissemination and re-use of open access materials and is preferred by many research funding bodies.

For further information about article processing charges, open access funding, and advice and support from Nature Research, please visit <https://www.nature.com/commspsychol/article-processing-charges>

At acceptance, you will be provided with instructions for completing this CC BY license on behalf of all authors. This grants us the necessary permissions to publish your paper. Additionally, you will be asked to declare that all required third party permissions have been obtained, and to provide billing

information in order to pay the article-processing charge (APC).

* **DATA AVAILABILITY:**

[Link redacted]

Best regards,

Marike

Marike Schiffer, PhD
Chief Editor
Communications Psychology

REVIEWERS' COMMENTS:

Reviewer #2 (Remarks to the Author):

The author has done a great job with revisions and I now fully support publication. I know the R&R process on this paper was a bit lengthy, but I think what was already a strong paper is now even further improved, and has indeed become one of the best and most interesting studies on this topic. I plan to assign this paper in my courses and hold it up as one of the best of the genre. Congratulations to the author. This is a great study.

Reviewer #3 (Remarks to the Author):

I thank the author for these final few revisions. I now fully support publication. Thank you for an excellent study and paper!